# Tilted Implants and Sinus Floor Elevation Techniques Compared in Posterior Edentulous Maxilla: A Retrospective Clinical Study over Four Years of Follow-Up

Enrico Felice Gherlone [1,2], Bianca D'Orto [1,2,*], Matteo Nagni [1,2], Paolo Capparè [1,2] and Raffaele Vinci [1,2]

1   Dental School, Vita-Salute San Raffaele University, 20132 Milan, Italy; gherlone.enrico@hsr.it (E.F.G.); nagnimatteo@libero.it (M.N.); cappare.paolo@hsr.it (P.C.); vinci.raffaele@hsr.it (R.V.)
2   Department of Dentistry, IRCCS San Raffaele Hospital, 20132 Milan, Italy
\*   Correspondence: b.dorto@libero.it

**Abstract:** The aim of this study was to evaluate the implant survival rate, marginal bone loss, and surgical and prosthetic complications of implants placed through sinus floor elevation and tilted implants engaged in basal bone to bypass the maxillary sinus. Sixty patients were enrolled for this study. According to the residual bone height of the posterior maxilla, the sample was divided into three groups of 20 patients: Group A (lateral sinus floor elevation), Group B (transcrestal sinus floor elevation), and Group C (tilted implants employed to bypass the sinus floor). Follow-up visits were performed one week after surgery, at three and six months, and then once a year for the next 4 years. The outcomes were the implant survival rate, marginal bone loss, and surgical and prosthetic complications. Although Groups A, B, and C demonstrated implant survival rates of 83.3%, 86.7%, and 98.3%, respectively, the statistical analysis showed no statistically significant difference between groups. Statistically significant differences between groups were also not found concerning marginal bone loss, as recorded by intra-oral X-ray measurements during follow-up examinations. Regarding complications, it was not possible to perform a statistical analysis. To reduce possible surgical risks, implant placement in basal bone could be preferred.

**Keywords:** posterior edentulous maxilla; maxillary sinus; sinus floor elevation; tilted implants

## 1. Introduction

Fixed rehabilitation of an atrophic maxilla may represent a real challenge for clinicians. Following tooth loss, the physiological process of bone resorption is combined with sinus pneumatization, which often impedes traditional implant placement in posterior sectors [1–3].

To allow patients to receive fixed rehabilitations, several therapeutic alternatives such as bone grafting and sinus lift techniques have been proposed to increase residual bone height; although these procedures have provided good long-term results [4,5], several complications, including Schneider's membrane perforation, grafted material infection and/or resorption, implant dislocation in the maxillary sinus, acute or chronic sinusitis, alveolo-antral artery injury, and benign paroxysmal vertigo, could occur [6,7].

To avoid these risks and the clinical time required, short [8] and tilted [9] implants could be considered a viable solution to engage basal bone.

In the choice between categories, considering the micromovements and peri-implant stresses and strains associated with ultra-short (5 mm length) implants [10,11], tilted implants or sinus floor elevation via a lateral approach is promoted in the case of severe bone atrophy (less than 5 mm) [12–15], and the transcrestal sinus lift technique is recommended if the residual bone height is at a minimum of 5 mm [16–18].

Although tilted implants might be the least risky choice [9], the following prerequisites should always be provided: adequate bone volume in the retrocanine area for implant placement at least 10 mm in length and combination with an axial implant [19,20].

In addition, implant placement in basal bone should always be considered in the presence of any conditions that could represent a possible contraindication to sinus augmentation, such as sinusitis, including allergic rhinitis, polyp, cyst, or tumor in the maxillary sinus, and a history of sinus surgery [21,22].

The aim of this retrospective clinical study was to evaluate and compare the implant survival rate (first outcome), marginal bone loss (second outcome), and surgical and prosthetic complications of implant prosthetic rehabilitation through implants placed via sinus floor elevation techniques (lateral approach and osteotome-mediated technique) and tilted implants engaged in basal bone to bypass the maxillary sinus.

## 2. Materials and Methods

### 2.1. Patient Selection

This retrospective study was performed at the Department of Dentistry, San Raffaele Hospital, Milan, Italy. The ethics committee approval number is 190/INT/2021.

The investigation was conducted according to the principles of the Declaration of Helsinki. STROBE (Strengthening the Reporting of Observational Studies in Epidemiology) guidelines were followed (http://www.strobe-statement.org/ (accessed 1 on April 2019)).

During the period from January 2015 to April 2019, patients with posterior edentulous maxilla (Applegate–Kennedy Class I, II, or III [23]) or severe impairment of residual teeth in the posterior maxilla were consecutively enrolled.

The eligibility criteria were as follows: patients over 18 years old with unilateral or bilateral partial edentulism of the maxilla, with residual bone height equal to or less than 6 mm or severe impairment of residual teeth in the posterior maxilla with maximum residual bone height of 6 mm after the healing period, and requiring fixed prosthetic rehabilitation to replace three or four teeth.

Patients with immunodeficiency, those with uncontrolled systemic diseases, those under bisphosphonate therapy or subjected to head and neck radiotherapy less than one year prior, those having severe malocclusion or parafunction, those unable to adhere to home and professional hygiene maintenance protocols, and smokers were excluded.

All diagnoses were made clinically and radiographically. The radiographic examination was conducted at the first level via panoramic radiography and at the second level via cone beam computed tomography (CBCT) to identify the residual bone height and whether the patient satisfied the inclusion criteria of the study (residual bone height of 6 mm or less). According to bone volume, the sample was divided in three groups (Table 1).

**Table 1.** Sample division according to residual bone height, bone volume in the retrocanine area, possibility or not of combining a tilted implant with an axial one, and presence or absence of any contraindication to sinus augmentation.

| | Group A (Sinus Floor Augmentation via Lateral Approach) | Group B (Transcrestal Sinus Floor Elevation) | Group C (One Tilted and One Axial Implants) |
|---|---|---|---|
| Residual bone height | Less than 5 mm, inadequate bone volume in the retrocanine area for tilted implant placement at least 10 mm long, impossibility of combining a tilted implant with an axial one and absence of any contraindication to sinus augmentation [15–19] | Minimum of 5 mm [20–22] | Less than 7 mm, adequate bone volume in the retrocanine area for tilted implant placement at least 10 mm long, possibility of combining a tilted implant with an axial one and contraindication to sinus augmentation [18–22] |

Written informed consent for implant prosthetic rehabilitation was obtained from all patients prior to the beginning of the study, and the local ethical committee approved the study; professional oral hygiene was provided before surgery.

### 2.2. Surgical Procedures

*Group A: Sinus floor elevation through a Lateral Window Technique*

All surgical procedures were performed by the same surgeon with advanced surgical experience. As for the other surgery types, one hour prior, patients received 2 g amoxicillin and clavulanic acid, and they received another 1 g twice a day for a week after the surgical procedure (clarithromycin was prescribed as an alternative in case of allergy, 2 g before surgery and 1 g twice a day for the following week).

Surgery was performed under anesthesia induced by local infiltration of opticain solution with adrenaline 1:80,000 (AstraZeneca, Milan, Italy). The same protocol was applied for all techniques.

The first incision was made on the top of the alveolar crest, shifted on the palatal side to obtain the same level of keratinized mucosa on both flap sides. Then, distal and mesial vertical release incisions were performed to expose the underlying bone crest. The full-thickness flap was elevated to preserve anatomical subperiosteal structures.

The flap was detached to expose the anterior piriform cortex and canine draft, used as landmarks, and to identify the maxillary sinus, often available in transparency from the lateral bone wall. A bony window was drawn using a sterile pencil on the lateral wall, behind the canine draft, according to the size and location of the maxillary sinus and the implant insertion site. Then, a high-speed handpiece with a diamond bur was employed to outline the antrostomy. A bone scraper was used to obtain autologous bone chips from the bony window. To preserve the Schneiderian membrane from injuries, a piezoelectric instrument was employed for bony window detachment.

The elevation degree was set according to the vertical defect's extension, proceeding from the inferior-medial sinus wall to the distal.

The implant sites were prepared using a lance-shaped drill followed by drills of increasing diameter; fixtures were then placed. The implants placed for this group belonged to the K line (Winsix, Biosafin, Ancona, Italy); this implant type has a cylindrical shape with a truncated conical body characterized by a self-threading coil with differentiated depth and thickness and variable geometry to modulate primary stability during surgery. The macromorphology is characterized by variable geometry, the coils gradually varying from square to triangular and varying in depth to favor vertical micro-expansion and progressive horizontal expansion; it has wide and deep unloading grooves for the deposition of bone chips and the formation of clots during the screwing phase. Due to the properties of these implants, they were chosen for both methods of sinus lift (lateral approach and transcrestal approach), by agreement of the surgeon and prosthetist.

The autologous bone graft obtained from the bony window was the only biomaterial applied, and it was placed around implants to promote bone regeneration [24].

An adsorbable hemostatic gelatin (Spongostan, Ethicon, Johnson & Johnson, New Brunswick, NJ, USA) was employed to promote clot formation and to contain the exposed sinus area following the removal of the bony opening that provided access to the cavity.

Flap adaptation and suturing were performed using 3–0 non-resorbable sutures (Vicryl; Ethicon, Johnson & Johnson, New Brunswick, NJ, USA).

*Group B: Sinus floor elevation through an Osteotome-Mediated Technique*

The first incision was made on the top of the alveolar crest, shifted on the palatal side to obtain the same level of keratinized mucosa on both sides of the flap. Distal and mesial vertical release incisions were performed to create a full-thickness flap, exposing the underlying bone crest and preserving anatomical epiperiosteal structures. A lance-shaped drill was employed for 2 mm to drill the cortical bone. A pilot drill of ø 2.00 was applied to create an implant insertion site and to define the fixture's setting. Then, osteotomes of progressively increasing diameter were gradually driven to sinus floor fracture and

Schneiderian membrane elevation. The diameter of the last osteotomy was less than the fixture diameter to promote primary mechanical stability. Any biomaterial was applied before implant placement. Only an adsorbable hemostatic gelatin (Spongostan, Ethicon, Johnson & Johnson, New Brunswick, NJ, USA) was employed to promote clot formation and to retain membrane elevation. Flap adaptation and suturing were performed using 3–0 non-resorbable sutures (Vicryl; Ethicon, Johnson & Johnson, New Brunswick, NJ, USA).

*Group C: Tilted implants*

The first incision was made on the top of the alveolar crest, shifted on the palatal side to obtain the same level of keratinized mucosa on both sides of the flap. Then, distal and mesial vertical release incisions were performed to expose the underlying bone crest.

The obtained full-thickness flap allowed us to preserve subperiosteal anatomical structures from injuries and to expose the canine draft and maxillary sinus, often available in transparency from the lateral bone wall. The tilted implant was placed first, in the vestibulo-palatal direction and adjacent to the mesial wall of the maxillary sinus; when possible, the apex of the fixtures engaged the inferior-medial wall of the inferior-distal cortical of the piriform opening.

Every tilted implant was associated with an axial implant, placed according to the traditional system in the canine or lateral incisor region. Straight implant placement always occurred after tilted implant insertion according to their position and angulation.

The implants placed for this group belonged to the TT line (Winsix, Biosafin, Ancona, Italy), which has a single implant body with a specific macromorphology to achieve maximum implant stability; it is also excellent for immature loading and differs in having either an internal hex (TTi) or an external hex (TTx). On a macromorphological level, they have double-threaded, double-principled coils for easy implant insertion with half the number of turns. The groove in the lower part of the loop decompresses the bone by dissipating forces and facilitates clot deposition. At the same time, it increases the implant surface by facilitating the neoformation of cells. The apex is conical and undersized by 1.3 to 1.8 with respect to the diameter of the implant; it is strongly tapered to obtain an osteotomic effect and to facilitate the inclined insertion of implants, even in the case of reduced bone availability.

Because of their characteristics and eligibility for immediate loading, they were selected for this group by agreement of the surgeon and prosthetist.

A lanceolate drill was employed to perforate cortical bone. A pilot drill of ø 2.00 was applied to create an implant insertion site and to define the fixture's setting. A positioning pin was plugged to verify the implant location, emergence, and angulation.

Drills of progressive diameter were employed up to the final fixture's diameter. The site was over-prepared vertically and sub-prepared transversely to promote primary mechanical stability. The implant neck was aimed to be positioned at bone level. The insertion torque ranging between 30 and 40 N·cm before final seating of the implant, allowing for immediate loading.

A manual screwer was applied when incomplete seating of the implant occurred, and bi-cortical anchorage was established whenever possible.

To compensate for the lack of parallelism between implants, angulated abutments (Extreme Abutment, EA® Winsix, Biosafin) at 30 degrees were screwed on tilted implants; straight abutments were screwed on axial implants. The flap was adapted around the structure. Suturing was performed using 3–0 non-resorbable sutures (Vicryl; Ethicon, Johnson & Johnson, New Brunswick, NJ, USA).

Post-Surgical Protocol

Immediately after surgery, intra-oral X-rays were performed to verify the correct implant position.

Antibiotic therapy (amoxicillin and clavulanic acid 1 g or clarithromycin 1 g in case of allergy, twice daily for 7 days after surgery) and analgesic therapy (non-steroidal anti-inflammatory drugs, as needed) were prescribed for each patient. Mouth rinsing with a

chlorhexidine-digluconate-containing solution (0.12% or 0.2%) was recommended twice daily for 10 days. One week after the surgical procedure, sutures were removed. The same post-surgical protocol was applied for all procedures.

### 2.3. Prosthetic Protocol

In both sinus floor elevation procedures (Group A and Group B), the implants were covered for about 4 months; then, reopening was performed, and cap screws were replaced with healing screws. An acrylic provisional prosthesis composed of three or four teeth according to the antagonist arch and the presence or absence of adjacent teeth was delivered to each patient. Screw access holes were covered with provisional resin (Fermit, Ivoclar Vivadent, Naturno, Bolzano, Italy). We performed the appropriate evaluation checks of the device, and after another four months, the provisional prothesis was replaced with a metal ceramic or resin implant-supported final prosthesis composed of three or four units.

Unlike these procedures, which involved a deferred load, in Group C, in accordance with several prior research results, patients were subjected to immediate loading [25,26].

One week before surgery, preliminary traditional impressions were taken to obtain an all-acrylic resin provisional prothesis composed of three or four teeth.

To enable manufacture of a high-density baked all-acrylic prosthesis with titanium cylinders, pickup impressions (Permadyne, ESPE, Seefeld, Germany) of the implants were made after suturing.

About 3 h after the surgery, a screw-retained acrylic provisional prosthesis with three or four teeth was delivered. Provisional resin (Fermit, Ivoclar Vivadent, Naturno) was used to cover screw access holes. Four months later, the provisional prothesis was replaced with a metal ceramic or resin implant-supported final prosthesis composed of three teeth.

In all groups, articulating papers (40 μm Bausch Articulating Paper) were applied to obtain central contacts made on all masticatory units (stating occlusion) in the provisional prothesis and dynamic occlusion, involving premolar guidance, definitively.

### 2.4. Follow-Up

Follow-up visits were performed 1 week after surgery, at 3 and 6 months, and then once a year for the next 4 years. Each patient was placed in a professional oral hygiene program that would allow both for limiting complications [27,28] and for monitoring and interception of any complications.

1. *Implant survival rate.* The implant survival rate was dependent on the number of implants lost during the follow-up period due to mobility associated with progressive marginal bone loss due to peri-implantitis. Implant loss was classified according to the period: if it occurred within 6 months of fixture placement, it was called early failure; after 6 months, it was called late failure. Early failure was usually intercepted at the reopening stage, when there was a lack of osseointegration of the implant. In the case of late failure, there were signs of peri-implantitis, implant mobility, radiolucent areas around fixtures, mucosal suppuration, and/or pain during the follow-up period.

2. *Marginal bone loss (MBL).* The MBL was evaluated via digital phosphor intra-oral radiography performed for each patient using the parallel cone technique at 6, 12, 24, 36, and 48 months. To assess marginal bone trends, measurements were performed only after image calibration. Digora 2.5 software (Soredex, Tuusula, Finland) was used as an analysis platform, making use of the specific measurement tool contained therein. As a first step, calibration (pixels/mm) of the instrument was performed, using the implant diameter of the survey site as the known unit. Next, any changes in the height of the peri-implant marginal bone in relation to the most coronal part of the implant and the point of contact between the implant and marginal ridge were measured. To evaluate bone resorption, a line passing over the shoulder of the implant was considered as a reference point for measurement from which a straight line was drawn parallel to the long axis of the implant to the most coronal point where the bone met the fixture both mesially and distally. The software automatically

provided, in relation to the calibration, the distance between the two points measured in millimeters. To reduce human error, this measurement was performed by three operators, and the average of the three measurements was considered. To evaluate the marginal bone level, first the mesial and distal measurements were taken, then the averages of the mesial, of the distal, and between the two values of a single implant site (MBL, marginal bone level) were calculated, as reported in Section 3. Marginal bone levels detected were divided into two categories according to the implant position, whether mesial (Implant 1/I1) or distal (Implant 2/I2). The first group included only axial implants; the second group also included tilted fixtures, always placed distally and in association with a mesial axial implant (Implant 1/I1). The data thus obtained were then statistically investigated.

3.  *Surgical complications.* Surgical complications were divided according to the surgical procedure.
4.  *Prosthetic complications.* These included fracture of the provisional prothesis, unscrewing of temporary crowns and/or abutments (Group C), unscrewing of final crowns and/or abutments (Group C), and chipping.

*2.5. Statistical Analysis*

Statistical analysis was performed for numerical parameters using SPSS for Windows version 18.0 (SPSS Inc., Chicago, IL, USA). Descriptive analysis was performed using the mean ± standard deviation.

The different implant survival rates between the surgical procedures, based on the number of implants lost in each group, were compared using the test of between-subject effects according to one-way analysis of variance (ANOVA).

Analysis of variance was used to investigate changes in the bone level over time. All statistical comparisons were conducted at the 0.05 significance level. The null hypothesis was that there would be no difference in mean marginal bone changes between implants. Regarding complications, due to the few cases observed, a statistical analysis could not be performed.

**3. Results**

According to the inclusion and exclusion criteria, 60 patients (32 males, 28 females) with an edentulous posterior maxilla (Applegate–Kennedy Class I, II, or III) or the need for avulsion of residual teeth in the posterior section were enrolled for this study. The mean age was 64 years (range: 52–76). The sample was divided into three groups of 20 according to the surgical procedure they received.

Every surgery involved the placement of two implants to support screw-retained prostheses of a minimum of three and a maximum of four dental units according to the antagonist arch (presence or absence of the lower first molar) and presence or absence of adjacent distal teeth. Fixtures were placed at sites 14 and 16, sites 14 and 15, or sites 14, 15, and 16 and in the same sites in the contralateral emi-arch.

A total of 144 dental implants (K or TTi or TTx, Winsix, Biosafin, Ancona, Italy) were placed. Group A received 48 implants, Group B received 46 implants, and Group C received 50 implants (Table 2).

Immediate loading was performed only in Group C; in both sinus floor elevation techniques, implant loading occurred approximately 4 months after implant placement.

1.  *Implant survival rate.* In the lateral sinus floor elevation technique (Group A), no implants were lost in the first six months after surgery; two fixtures were lost in the following period. In the transcrestal approach (Group B), one implant was lost in the first six months after surgery, and only one was lost later. Only one tilted implant (Group C) was lost early; no implants were lost in the following period.

Group A, Group B, and Group C demonstrated implant survival rates of 95.83%, 95.65%, and 98%, respectively (Table 3).

**Table 2.** Number, diameter, and length of dental implants classified by group.

| | | Dental Implant Details | | | |
|---|---|---|---|---|---|
| | | Length 9 mm | Length 11 mm | Length 13 mm | Length 15 mm |
| **Group A** (sinus floor augmentation via lateral approach) n = 48 | diameter 3.3 mm | 6 | 7 | 0 | 0 |
| | diameter 3.8 mm | 29 | 6 | 0 | 0 |
| **Group B** (transcrestal sinus floor elevation) n = 46 | diameter 3.3 mm | 2 | 3 | 1 | 0 |
| | diameter 3.8 mm | 16 | 21 | 3 | 0 |
| **Group C** (one tilted and one axial implant) n = 50 | diameter 3.3 mm | 0 | 0 | 4 | 4 |
| | diameter 3.8 mm | 0 | 2 | 29 | 11 |

**Table 3.** Implant failure before or after the osseointegration period (6 months) and implant survival rates according to the surgical procedure at the end of the follow-up period (4 years).

| | Implants Placed | Early Failure | Late Failure | Implant Survival Rate |
|---|---|---|---|---|
| Group A | 48 | 0 | 2 | 95.83% |
| Group B | 46 | 1 | 1 | 95.65% |
| Group C | 50 | 1 | 0 | 98% |

However, a one-way analysis of variance (ANOVA) revealed no differences among groups in terms of the proportion of lost implants ($F_{(2, 60)} = 0.54$, $p = 0.59$, n.s.). Although seemingly different from one another, the estimated mean values did not differ statistically (Table 4).

**Table 4.** Differences among groups in terms of the proportion of lost implants.

| Dependent Variable: Prop_lost Dental Implants | | | | | |
|---|---|---|---|---|---|
| **Source** | **Type III Sum of Squares** | **Df** | **Mean Square** | **F** | **Sig.** |
| **Corrected Model** | 0.036 [a] | 2 | 0.018 | 0.539 | 0.586 |
| **Intercept** | 0.363 | 1 | 0.363 | 10.775 | 0.002 |
| **Group** | 0.036 | 2 | 0.018 | 0.539 | 0.586 |
| **Error** | 1.920 | 57 | 0.034 | | |
| **Total** | 2.319 | 60 | | | |
| **Corrected Total** | 1.956 | 59 | | | |

[a] R Squared = 0.019 (Adjusted R Squared = −0.016).

2. *Marginal Bone Loss.* Statistical analysis was also performed for marginal bone loss, evaluated 6 months after the surgical procedure, 12 months after the surgical procedure, and once a year subsequently. The values obtained were divided into two categories according to the fixture position (Tables 5 and 6).

**Table 5.** Average marginal bone loss (millimeters) of mesial implants (I1) observed during follow-up.

| Descriptive Statistics | | | | |
|---|---|---|---|---|
| | **Group** | **Mean** | **Std. Deviation** | **N** |
| I1_MBL 6 months (mm) | A | 0.970 | 0.1455 | 20 |
| | B | 0.915 | 0.1387 | 20 |
| | C | 0.920 | 0.1508 | 20 |
| | Total | 0.935 | 0.1448 | 60 |
| I1_MBL 12 months (mm) | A | 1.095 | 0.1356 | 20 |
| | B | 1.085 | 0.1663 | 20 |
| | C | 1.040 | 0.1847 | 20 |
| | Total | 1.073 | 0.1625 | 60 |

**Table 5.** *Cont.*

| | Group | Mean | Std. Deviation | N |
|---|---|---|---|---|
| | | **Descriptive Statistics** | | |
| I1_MBL 24 months (mm) | A | 1.250 | 0.1235 | 20 |
| | B | 1.260 | 0.1729 | 20 |
| | C | 1.255 | 0.1572 | 20 |
| | Total | 1.255 | 0.1501 | 60 |
| I1_MBL 36 months (mm) | A | 1.470 | 0.0801 | 20 |
| | B | 1.500 | 0.1338 | 20 |
| | C | 1.475 | 0.0786 | 20 |
| | Total | 1.482 | 0.1000 | 60 |
| I1_MBL 48 months (mm) | A | 1.695 | 0.1986 | 20 |
| | B | 1.720 | 0.2238 | 20 |
| | C | 1.585 | 0.0933 | 20 |
| | Total | 1.667 | 0.1875 | 60 |

**Table 6.** Average marginal bone loss (millimeters) of distal implants (I2) observed during follow-up.

| | Group | Mean | Std. Deviation | N |
|---|---|---|---|---|
| | | **Descriptive Statistics** | | |
| I2_MBL 6 months (mm) | A | 0.918 | 0.1131 | 17 |
| | B | 0.888 | 0.1310 | 16 |
| | C | 0.936 | 0.1447 | 14 |
| | Total | 0.913 | 0.1279 | 47 |
| I2_MBL 12 months (mm) | A | 1.094 | 0.1249 | 17 |
| | B | 1.088 | 0.1708 | 16 |
| | C | 1.100 | 0.1177 | 14 |
| | Total | 1.094 | 0.1374 | 47 |
| I2_MBL 24 months (mm) | A | 1.306 | 0.1345 | 17 |
| | B | 1.238 | 0.1258 | 16 |
| | C | 1.236 | 0.1336 | 14 |
| | Total | 1.262 | 0.1328 | 47 |
| I2_MBL 36 months (mm) | A | 1.488 | 0.0781 | 17 |
| | B | 1.481 | 0.1109 | 16 |
| | C | 1.450 | 0.1092 | 14 |
| | Total | 1.474 | 0.0988 | 47 |
| I2_MBL 48 months (mm) | A | 1.588 | 0.0857 | 17 |
| | B | 1.681 | 0.1328 | 16 |
| | C | 1.600 | 0.1569 | 14 |
| | Total | 1.623 | 0.1306 | 47 |

Regarding Implant 1, as shown in Figure 1, a 3 (groups) × 5 (time) MANOVA revealed a main effect of time ($F_{(1, 57)} = 786.11$, $p < 0.001$), while other effects did not reach the conventional threshold of statistical significance. In other words, the MBL for Implant 1 tended to increase over the five time periods, regardless of the surgical approach (i.e., group) (Figure 1).

Regarding Implant 2, as shown in Figure 2, a 3 (groups) × 5 (time) MANOVA revealed a main effect of time ($F_{(1, 44)} = 680.31$, $p < 0.001$), while other effects did not reach the conventional threshold of statistical significance. In other words, the MBL for Implant 2 tended to increase over the five time periods, regardless of the surgical approach (i.e., group) (Figure 2).

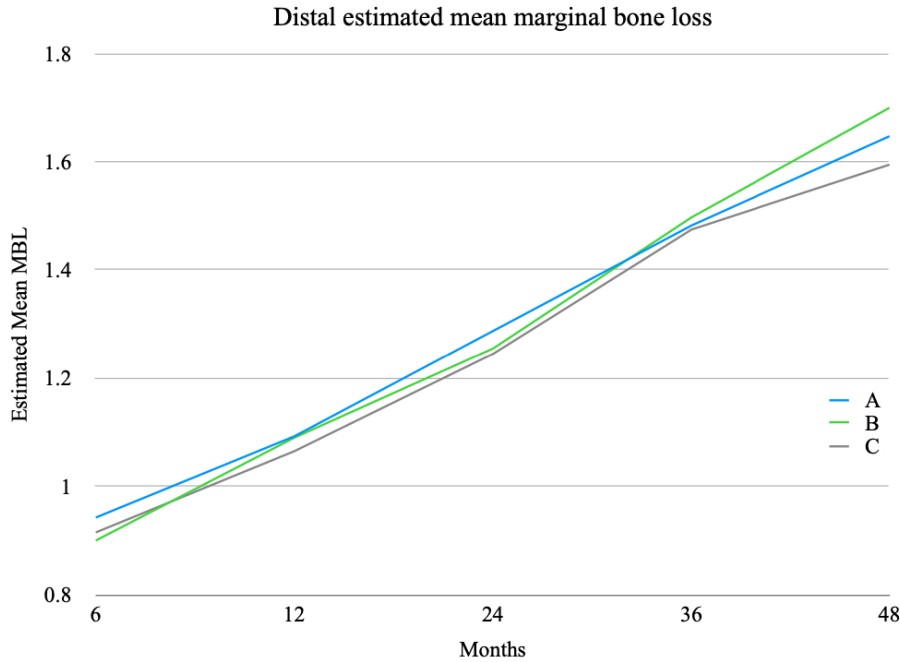

**Figure 1.** MBL for Implant 1, which tended to increase over the five time periods, irrespective of surgical approach.

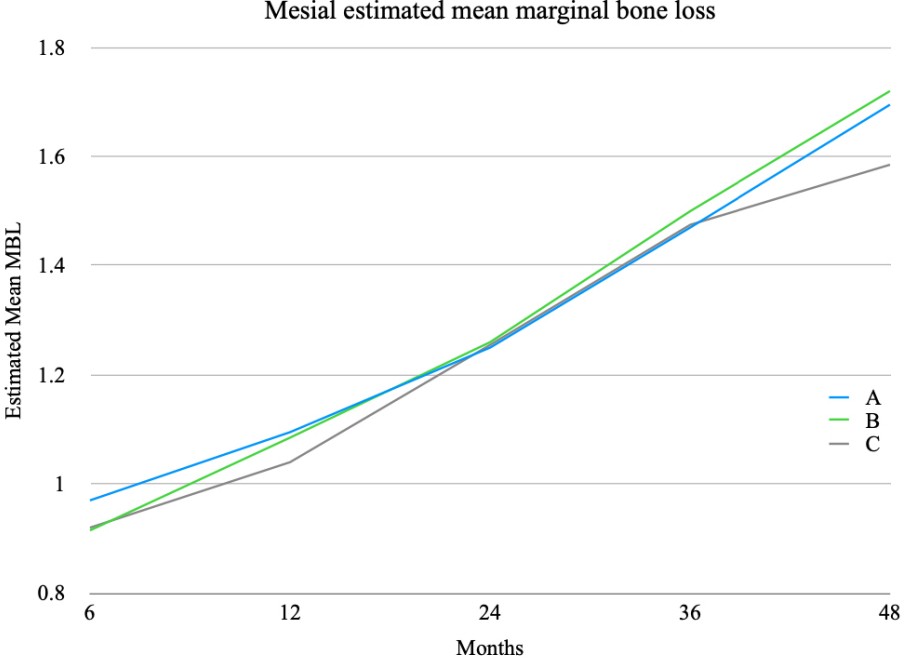

**Figure 2.** MBL for Implant 2, which tended to increase over the five time periods, irrespective of surgical approach.

3. *Surgical Complications.* All recorded complications were related to the lateral sinus floor elevation technique (Group A) or transcrestal sinus floor elevation (Group B). In Group C, there were no intra-operative complications. Three membrane perforations were reported in Group A. The complication was resolved intra-operatively by further detaching the Schneider membrane from the inferior-medial region to reposition the hole under the bone wall. This avoided leakage of the graft material and possible subsequent infection. In the same group, no other complications were reported. In Group B, the only problem encountered was paroxysmal benign positional vertigo

(PPBV), associated with the percussive action induced by the surgical mallet. After about one month, the complication resolved itself in all four cases where it was found.

4.  *Prosthetic Complications.* No prosthetic complications were reported during the follow-up period.

## 4. Discussion

In this retrospective clinical study, three different surgical techniques for the rehabilitation of a posterior maxilla with insufficient residual bone height for the placement of axial implants were compared.

Sinus lift techniques have been extensively discussed, and several authors have reported good short- and long-term results on implant survival rate.

Concerning a lateral sinus lift, Canullo et al., in their multicenter prospective study at 2 years of follow-up, reported an implant survival rate of 97% in patients with residual bone height between 1 and 4 mm who were treated with a lateral sinus augmentation using a nano-crystalline hydroxyapatite sole bone filler, simultaneous implant placement, and a deferred loading protocol [29].

Similar results were obtained by Schmitt et al. in their retrospective clinical study at 10 years of follow-up, in which they reported an implant survival rate of 95.45% in patients undergoing sinus lift via a lateral approach using autologous bone graft, implant placement after a four-month healing period, and a deferred loading protocol [30].

These results are similar to those obtained in this study, where the implant survival rate of fixtures placed by sinus lift via a lateral approach was 95.83%.

Beretta et al. [31], in their retrospective clinical study at 15 years of follow-up, compared implant survival in patients subjected to sinus lift with that in patients subjected to a lateral approach, depending on the implant placement protocol and biomaterial used. Implants placed at the same time as the sinus lift (residual bone height above 4 mm) provided similar results to implants placed after the healing period; autologous bone, according to other studies [32], provided better results than a heterologous bone graft. Although autologous bone is currently considered the gold standard in bone regeneration [33], good medium- and long-term results have been obtained in both sinus lift techniques even without bone grafting [34].

Considering these results, in the present study, autologous bone chips obtained from the lateral bone window were applied as the only bone regeneration material.

Concerning transcrestal sinus lift, Bruschi et al., in their retrospective clinical study at 10.43 ± 5.01 years (ranging from 5 to 16 years) of follow-up, reported a survival rate of 95.45% [35].

Similar results were obtained by Qian et al., in their randomized controlled trial at 10 years of follow-up, in which they reported an implant survival rate of 90.7% in the case of osteotome sinus floor elevation with deproteinized bovine bone mineral and 95.0% without bone grafting [36].

According to these authors, the transcrestal sinus lift, applied when residual bone height was at least 5 mm, was performed through an osteotome-mediated technique and without biomaterials, recording an implant survival rate of 95.65%.

Concerning tilted implants applied to bypass the sinus floor, the implant survival rate recorded in our study (98%) could be compared with those in other studies.

Aparicio et al., in their retrospective clinical study at 5 years of follow-up, reported an implant survival rate of 95.2% in immediate-loading rehabilitations of posterior edentulous maxilla with the placement of one axial and one tilted implant, concluding that tilted implants, longer than traditional ones, could increase the implant-to-bone contact area, promoting primary stability, reducing the prosthetic cantilever, and engaging basal bone [37].

Similar results were obtained by Fortin et al. [38] and Pozzi et al. [39], who reported implant survival rates of 100% at 5 years of follow-up and 96.3% at 3 years of follow-up, respectively, in the absence of intra- or post-operative surgical complications.

Regarding marginal bone loss, as reported by Antonoglou et al. in their systematic review and meta-analysis of clinical trials [40] and as confirmed by our results, implants placed via sinus lift techniques show values similar to those of implants placed via traditional methods.

The rationale for choosing between the different surgical procedures considered in this study could therefore be related to possible complications of sinus augmentation techniques, such as perforation of Schneider's membrane, graft infection, implant or graft dislocation in the maxillary sinus, acute or chronic sinusitis, injury of the alveolus-antral artery, and benign paroxysmal vertigo, which could occur [41,42].

In our study, the only recorded complication in Group A was Schneiderian membrane perforation, which is considered a prevalent occurrence in sinus floor augmentation via a lateral approach [43]; a similar situation occurred in Group B, where the only complication was paroxysmal benign positional vertigo [44].

According to several authors and the results of this study, where no complications were recorded in Group C, when possible, tilted implants could be proposed as a possible alternative in the rehabilitation of partially or totally edentulous maxilla [45–47], avoiding more invasive techniques and allowing immediate loading [9,37,45].

Regardless of possible complications, studying the cone beam CT scan before proceeding with any type of surgical approach may be crucial to evaluate the residual bone height and maxillary sinus conformation and, subsequently, to make the most appropriate surgical choice [15–22,48].

Furthermore, once the surgical technique that we consider most appropriate according to the above parameters has been chosen, pre-surgical planning is a valuable confirmation aid and an indication of how to perform surgery in a predictable way [49–51].

Since there may be discrepancies between the pre-surgical planning obtained from the cone beam CT scan and the clinical procedure [52,53], the surgeon's experience may be crucial in managing these variables and possible complications [54,55].

Also, in our case, the choice of an experienced surgeon supported by the aid of second-level radiographic examinations and pre-surgical planning may have had a positive influence on the obtained results.

However, for a more in-depth analysis, multicenter clinical studies with more variables, such as surgeon, prosthetist, and registrar, could be useful to compare the different techniques, also obtaining a larger sample of patients.

The choice of surgical procedure between sinus lift with a lateral or transcrestal approach and tilted implants may depend on several criteria such as the residual bone height, maxillary sinus conformation, and presence of any pathology affecting the maxillary sinus; as with any other surgical procedure, a risk–benefit ratio assessment could be crucial [12–14].

Although the results of the present study did not show any statistically significant differences between the groups, the finding that sinus lift techniques might involve a higher inherent risk of complications could be considered as one of the parameters of treatment choice [7].

## 5. Conclusions

Within the limitations of the present study, the obtained results suggest tilted implants as a possible alternative to sinus floor elevation procedures. Although there were no statistically significant differences in implant survival and marginal bone loss between the groups, tilted implants placed in the available bone presented fewer complications compared to sinus elevation via a lateral window approach or osteotome-mediated technique. It is possible to perform immediate partial rehabilitation over maxillary tilted implants with minimal complications. However, the surgical qualifications of the clinician may be crucial to performing all the listed procedures correctly. Further studies with enlarged samples may be necessary to confirm the obtained results.

**Author Contributions:** Conceptualization, E.F.G. and B.D.; methodology, P.C.; validation, R.V., P.C. and B.D.; formal analysis, M.N.; investigation, R.V.; resources, M.N.; data curation, B.D.; writing—original draft preparation, B.D.; writing—review and editing, M.N.; visualization, P.C.; supervision, E.F.G.; project administration, E.F.G. All authors have read and agreed to the published version of the manuscript.

**Funding:** This research received no external funding.

**Institutional Review Board Statement:** The study was conducted according to the guidelines of the Declaration of Helsinki and approved by the Institutional Review Board (or Ethics Committee) of Vita-Salute University, Milan, Italy.

**Informed Consent Statement:** Informed consent was obtained from all subjects involved in the study.

**Data Availability Statement:** The data presented in this study are available on reasonable request from the corresponding author.

**Conflicts of Interest:** The authors declare no conflict of interest.

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
