# Peer review of "Tilted Implants and Sinus Floor Elevation Techniques Compared in Posterior Edentulous Maxilla: A Retrospective Clinical Study over Four Years of Follow-Up"

_applsci, doi:10.3390/app12136729_

Round 1
Reviewer 1 Report
This manuscript presents an interesting study on survival rate, marginal bone loss, surgical and prosthetic complications of implants placed using sinus lift technics (lateral/crestal approaches) and tilted placement. The results showed that there are no significant differences between three groups. While not entirely novel, these results extend our understanding of the short- and long-term outcomes of these three procedures. However, I have some questions and comments.
#1. Even though the experimental period was 2015-2019, the approval date of the ethics committee seems to be 2021.
#2. Please provide an x-ray sample or schema explaining tilted placement of distal dental implant. In page 4, line 172, the authors state “bi-cortical anchorage”. Does the tilted distal implant mean zygomatic implant?
#3. In page4, line 148, the authors explain that “The diameter of the last osteotomy was equal to the fixture diameter”. How to gain the initial stabilization of dental implants in the lifted sinus floor site?
#4. Please provide the information of grafted bone (autologous only or with bone substitutes) in the sinus floor of the group A.
#5. In page 6, line 275, “early lost” of a tilted implant in group C was lost in the first 6 months?
#6. In Table 3, does “Early failure” mean lost before loading? In group C, was early failure immediate loaded implant?
#7 Please discuss the reason of implant failure in each group respectively.
Author Response
#1. Even though the experimental period was 2015-2019, the approval date of the ethics committee seems to be 2021.
Thank you for your suggestion. However, the study was retrospective, so we ask the ethical committee the present year 2022.
#2. Please provide an x-ray sample or schema explaining tilted placement of distal dental implant. In page 4, line 172, the authors state “bi-cortical anchorage”. Does the tilted distal implant mean zygomatic implant?
In the Materials and Methods section dedicated to group C, we have added a more detailed description of the placement of tilted implants. We remain available for further clarification.
#3. In page4, line 148, the authors explain that “The diameter of the last osteotomy was equal to the fixture diameter”. How to gain the initial stabilization of dental implants in the lifted sinus floor site?
We have corrected the sentence. To promote primary mechanical stability, transverse under-preparation of the implant site has always been performed.
#4. Please provide the information of grafted bone (autologous only or with bone substitutes) in the sinus floor of the group A.
Following your corrections, we have specified that the autologous bone graft obtained from the bone window was the only biomaterial applied.
#5. In page 6, line 275, “early lost” of a tilted implant in group C was lost in the first 6 months?
#6. In Table 3, does “Early failure” mean lost before loading? In group C, was early failure immediate loaded implant?
#7 Please discuss the reason of implant failure in each group respectively.
Concerning questions 5, 6 and 7, further specifications about implants loss and failure classification according to the timing from surgery have been included in “Materials and methods” section. We remain available for further clarification.
Reviewer 2 Report
L 39: Please start by explaining the conventional sinus bone augmentation procedures then their potential complications then the alternative that are the tilted implants to overcome the complications
L 46 and/or
L.85: year
L. 91: redundancy: less than 6 mm
L. 133: Antibiotic regimen description: redundant
L. 206: Implant survival rate criteria are unclear as they seem to correspond to success rate. Please clarify it. No reference was added to justify this choice. Radiolucency around fixture should be better delimited as the initial bone remodeling will create radiolucency.
L. 348: et al
Result section:
- Please specify each group corresponds to what kind of surgery and add this to the legend of table 2
- For the survival rate it was calculated for all groups at 48 months? This should be indicated in the table 3
Discussion:
The discussion is insufficient as it cites data from the literature about survival rates of implants placed in augmented sinus and no discussion about the marginal bone loss (one sentence from line 390-391) observed for the different groups in this study and no comparison to published data about this point
The use of short implants and the potential advantage of tilted placement should be addressed
Author Response
Some concepts are repeated within the introduction, and I would recommend avoiding repeating concepts and making an introduction easier to read. It looks sort of untidy.
According with your suggestions we have tried to modify the introduction trying to keep the focus on the concept of using the basal bone whenever possible.
MATERIAL AND METHODS
Much information is missing, such as:
Type of implants?? The number of patients? The number of implants? How many patients/implants for each study group? How were they assigned to each study group?
The type of implant was added to the results. The choice between the various types of implants proposed by the company depended on the guidelines they provided, as is our custom. The number of patients and the number of implants was written down as well as the sample classification system, summarized in Table 1 and in accordance with the literature results we cited in the same table.
The lateral window technique does not specify what type of bone was grafted and if collagen membrane was used to close the window.
We added several specifications in Materials and Methods section:
“Autologous bone graft obtained from bony window was the only biomaterials applied and it was placed around implants to promote bone regeneration [24].
An adsorbable hemostatic gelatin (Spongostan, Ethicon, Johnson & Johnson, New Brunswick, NJ, USA) was employed to promote clot formation and to contain the exposed sinus area following the removal of the bony opening that provided access to the cavity.”
In the osteotome technique, it is not specified if there were any biomaterial grafted, and the method used for elevation is not fully explained.
We added several specifications in Materials and Methods section:
“Any biomaterial was applied before implants placement. Only an adsorbable hemostatic gelatin (Spongostan, Ethicon, Johnson & Johnson, New Brunswick, NJ, USA) was employed to promote clot formation and to retain membrane elevation.”
When explaining the used prosthetic protocol, it does not say a word of the protocol used for provisional prostheses manufactured in the closed/open sinus elevation group. The immediate loading group does not outline the type of occlusion of the immediate prosthesis. Was there any function with the prosthesis? And in the eccentrics?
Further specifications regarding the characteristics of the provisional prosthesis in groups A and B have been added.
Information on the occlusion of provisional and definitive prostheses has been added.
DISCUSSION
It is important to outline how patients were assigned to each group. Suppose they were designated as explained in the material and methods. In that case, the difference of the implant outcome and the marginal bone loss could be explained by the initial situation of the patient, and I consider this fact should be discussed.
Placing tilted implants is not easy at all, and it is easy to introduce the osteotomy in the sinus or incline the implant more than necessary. I think that guided surgery should be mentioned for this aspect, so dentists with little experience with tilted implants can safely make them.
Limitations of the study should be discussed in the discussion. The sample size is way too small because very similar results are expected to be found among the study groups. Considerable sample size would be needed to find statistical differences. Other possible limitations should be described to make the manuscript more valuable.
All surgical procedures were performed by an experienced operator, Professor Raffaele Vinci, maxillofacial surgeon and Director of the School of Specialization in Oral Surgery at our department (listed among the authors). It is implied that the procedures listed require advanced surgical preparation. The aim is to understand if, when possible, it is preferable to exploit the basal bone rather than using procedures that are no less effective but could cause more complications.
The main limitations of the study were added in the conclusion.
Reviewer 3 Report
INTRODUCTION
Some concepts are repeated within the introduction, and I would recommend avoiding repeating concepts and making an introduction easier to read. It looks sort of untidy.
MATERIAL AND METHODS
Much information is missing, such as:
Type of implants?? The number of patients? The number of implants? How many patients/implants for each study group? How were they assigned to each study group?
The lateral window technique does not specify what type of bone was grafted and if collagen membrane was used to close the window.
In the osteotome technique, it is not specified if there were any biomaterial grafted, and the method used for elevation is not fully explained
When explaining the used prosthetic protocol, it does not say a word of the protocol used for provisional prostheses manufactured in the closed/open sinus elevation group. The immediate loading group does not outline the type of occlusion of the immediate prosthesis. Was there any function with the prosthesis? And in the eccentrics?
DISCUSSION
It is important to outline how patients were assigned to each group. Suppose they were designated as explained in the material and methods. In that case, the difference of the implant outcome and the marginal bone loss could be explained by the initial situation of the patient, and I consider this fact should be discussed.
Placing tilted implants is not easy at all, and it is easy to introduce the osteotomy in the sinus or incline the implant more than necessary. I think that guided surgery should be mentioned for this aspect, so dentists with little experience with tilted implants can safely make them.
Limitations of the study should be discussed in the discussion. The sample size is way too small because very similar results are expected to be found among the study groups. Considerable sample size would be needed to find statistical differences. Other possible limitations should be described to make the manuscript more valuable.
Author Response

(The authors gave the same response as above.)

Round 2
Reviewer 1 Report
Table 1 is difficult to see. Please correct it.
Author Response
Thank you for your suggestion; we have reversed lines and columns to clarify the table content.
Reviewer 2 Report
Unfortunately I didn't receive any specific response sheet for my comments and report.
Author Response
Dear reviewer, according to your suggestions:
L 39: Please start by explaining the conventional sinus bone augmentation procedures then their potential complications then the alternatives that are the tilted implants to overcome the complications.
The possible complications of sinus lift have been listed so as to bring out the risks associated with this procedure and promote the placement of implants in basal bone.
L 46 and/or
L.85: year
- 91: redundancy: less than 6 mm
- 133: Antibiotic regimen description: redundant
We have modified the above sections as requested.
- 206: Implant survival rate criteria are unclear as they seem to correspond to success rate. Please clarify it. No reference was added to justify this choice. Radiolucency around fixture should be better delimited as the initial bone remodeling will create radiolucency.
We have corrected the section to make it clearer.
Result section:
Please specify each group corresponds to what kind of surgery and add this to the legend of table 2.
We have added the kind of surgical procedure related to respective groups in order to make the table more explicit.
For the survival rate it was calculated for all groups at 48 months? This should be indicated in the table 3.
We specified that implant survival was assessed at the end of the follow-up period in the table caption.
Discussion:
The discussion is insufficient as it cites data from the literature about survival rates of implants placed in augmented sinus and no discussion about the marginal bone loss (one sentence from line 390 - 391) observed for the different groups in this study and no comparison to published data about this point.
The use of short implants and the potential advantage of tilted placement should be addressed.
Further studies and considerations were added to the discussion thus. With regard to short implants, a section was included in the introduction; we did not go into the discussion because they were not the subject of our study.
Reviewer 3 Report
I feel that type of implant must be decided by the surgeon and the prosthodontist. The type of implant used for each indication in the manuscript should be written, and the decision on whether to use one or another should be discussed in the discussion. To me, the answer you provide me is not enough. The implant company is not responsible for the patient. The surgeon and the prosthodontist are the only ones responsible for the patient, and they are who must decide the type of implant to use in any indication. It would be fascinating for the readership to know the parameters the clinicians used when choosing the type of implant.
I do not doubt that an outstanding clinician placed implants, and indeed, I think that placing tilted implants is challenging and must be performed by a highly-skilled surgeon. However, nowadays, many valuable tools are available to make surgeries easier and safer for both the surgeon and the patient, and I feel that it must be discussed.
I see a significant flaw when deciding the type of treatment. The choice is based on the amount of residual bone in the patient, which may be an important flaw when comparing the outcomes of the technique. It is not the same to have more available bone to place tilted implants that to have to make the body make new bone when doing osteotomes because less bone is available. This possible flaw is a fact that must be discussed in the manuscript.
The main limitations must be discussed in the discussion and not at the conclusion. The discussion is the part of the document where researchers can digest all information got during the research, and it is precious for the readership. Please, discuss all your results and the problems you had during the investigation so that everybody can learn the most from your experience.
Author Response
According to your considerations, we have added to the discussion the importance of CBCT in the assessment of residual bone height and the choice of the most suitable treatment for the patient, also based on anatomical variables such as sinus conformation.
For the rest, regarding the choice of treatment, we obtained the literature, as shown in table 1.
From our point of view, the association between literature, clinical experience of the operator, and the aid of level II radiographic examinations could be the key in choosing a particular surgical approach.
We also specified that there may be discrepancies between CBCT and clinical reality and that the help of an experienced surgeon could positively influence the results.
Furthermore, precisely on the basis of what we reported, we emphasised that it would be important to perform one or more multicentre clinical studies in order to enlarge the sample and have more realistic results.
We thank you for your suggestions and remain at your disposal for any further modifications, hoping to have fulfilled your previous requests.
Round 3
Reviewer 3 Report
I think the manuscript has improved a lot. However, I still think that comparing the success of different techniques that have been chosen according to various clinical situations is a significant bias that has not been yet addressed in the discussion. No statistical differences were seen among the studied techniques, but I feel this is because the sample size, with such a small data dispersion, must be much higher, and not because of the used technique. Maybe it would be interesting to discuss this in the discussion section. All other issues have been ideally addressed. Thank you very much.
Author Response
Dear reviewer,
thank you again for your suggestions.
We agree with you on the difficulty of comparing the success of different techniques. The aim is to assess the risk-benefit ratio of different surgical procedures in the most appropriate therapeutic choice for the individual.
Based on your suggestion, we have added a section in the discussion to emphasize your proposed concept.
We thank you for your kindness and helpfulness and we remain at your disposal for any further improvements.
